# High-Performance Catalytic Wet Oxidation of Excess Activated Sludge Derived from Pharmaceutical Wastewater Treatment Process over a Cu/γ-Al₂O₃ Catalyst

**Shangye Chu [1], Xu Zeng [2,*], Hai Lin [1,*] and Yuting Zhu [2]**

[1] School of Energy and Environmental Engineering, University of Science and Technology Beijing, Beijing 100083, China; chushangye@163.com

[2] College of Environmental Science and Engineering, Tongji University, Shanghai 200092, China; 15021248600@126.com

* Correspondence: zengxu@tongji.edu.cn (X.Z.); linhai@ces.ustb.edu.cn (H.L.)

**Abstract:** The performance of catalytic wet oxidation of excess activated sludge derived from pharmaceutical wastewater treatment over a Cu/γ-Al₂O₃ catalyst was investigated. The experiments were performed with a stainless steel autoclave reactor by using the prepared Cu/γ-Al₂O₃ catalyst. The effects of reaction conditions were examined, including additional catalyst amount, reaction temperature, time, and initial oxygen pressure. Results demonstrated that the catalyst, fabricated via co-precipitating method, has excellent catalytic performance. Through the study on condition optimization, the highest removal rates of volatile suspended solids (VSS, 93.6%) and total chemical oxygen demand (TCOD, 76.5%) were acquired with the reaction temperature 260 °C, time 60 min, initial oxygen pressure 1.0 MPa, and the prepared catalyst 5.0 g/L. The volatile fatty acids (VFAs) produced from the wet oxidation of sludge included acetic (mainly), propanoic, isobutyric, and isovaleric acids, which have great potential for the utilization as organic carbon sources in biological wastewater treatment plant. These results demonstrated that the proposed method, catalytic wet oxidation over Cu/γ-Al₂O₃ catalyst, is effective for treating excess activated sludge and resource utilization of organic carbon in the sludge.

**Keywords:** catalytic wet oxidation; pharmaceutical sludge; copper catalyst; carbon source

## 1. Introduction

During the past few decades, the detection of pharmaceuticals in the aquatic environment has attracted much attention. Extensive studies have been performed on the environmental risk of these pharmaceuticals for human beings. In particular, antibiotics cause great harm even at low concentrations in the environment. On the one hand, some pharmaceuticals enter the aquatic environment through the municipal wastewater treatment process. On the other hand, pharmaceutical sludge is another source of pharmaceuticals entering the aquatic environment. It should be noted that large amounts of excess activated sludge were produced from the wastewater treatment plant, such as the treatment of pharmaceutical, chemical, electroplating, printing, and dyeing wastewater. The sludge always includes hazardous and refractory organic pollutants and heavy metals [1]. In particular, certain pharmaceutical sludge has a higher environmental risk because of hazardous antibiotics, such as florfenicol, sulfonamide, benzylpenicillin, aureomycin, etc. [2,3]. Biotechnology and incineration are the common methods for treating the sludge. Biotechnology was adopted at a lower cost. However, the treatment efficiency was also lower because the pollutants in the sludge, including organic and inorganic materials, such as heavy metals, are always hazardous for the survival of microorganisms. Incineration is always used for sludge disposal. However, the cost was always very high, and it can produce secondary pollution, such as dioxin and nitrogen oxides. Landfill and composting methods are seldom

used for the treatment of pharmaceutical sludge nowadays. Therefore, suitable treatment methods have become a growing demand. It should be noted that energy and resource utilization from the sludge has attracted more and more attention in recent years. The treatment process would be more fascinating once the sludge could be utilized as a resource or a source of value-added chemicals, which were produced simultaneously when the sludge was disposed.

In recent years, advanced oxidation processes (AOPs) have always been adopted to improve the removal of pharmaceuticals during wastewater treatment, which can remove organic contaminants via oxidation through the generated hydroxyl radicals. The hydroxyl radicals are strong oxidants known to be effective for the degradation of a wide range of organic pollutants. For example, the heterogeneous photocatalysis for the photodegradation of tartrazine yellow dye and other pollutants has been reported, which showed excellent degradation results [4,5]. Among the diverse AOPs, wet oxidation has attracted much attention. Recently, wet oxidation technology has been assumed to be an effective treatment method for hazardous organic wastewater and sludge, which was conducted under high temperature and high pressure [6]. Thermal hydrolysis happened firstly, which induced the liquefaction of sludge, i.e., the dissolution of a solid into a liquid. In hydrolysis, the volatile suspended solids are transferred to the liquid phase and decompose into small materials. Subsequently, free radical agents generated under hydrothermal conditions oxidized the organic pollutants in the liquid phase, producing small molecule organics, even $CO_2$ and $H_2O$ [7]. Extensive studies proved that the major intermediate products include organic carboxylic acids, mainly acetic acid [8]. In Strong's study, the organic acids were produced from the destruction of municipal sludge using a wet oxidation process [9]. The results demonstrated that the organic carbon in the sludge could be converted as an organic carbon source for wastewater treatment. It is very useful that the exothermic wet oxidation could provide energy for the self-maintenance of the reaction system. Furthermore, the process avoids the demand for the energy-intensive dewatering step. Therefore, the wet oxidation technology would be utilized as an ideal method. Furthermore, catalyst addition is an efficient method to enhance the reaction efficiency and decrease the operating conditions [10,11]. During the past decades, extensive studies have been performed on the catalysis of transition metals due to their excellent oxidizing characteristics [12,13]. Among these catalysts, Cu-based catalysts have attracted considerable attention. It has been proved that a Cu-loaded catalyst can enhance the oxidation of acetaldehyde under wet oxidation conditions [14]. A Cu-based catalyst exhibited high efficiency in the oxidation of aromatic compounds in Chou's study [15]. Taran et al. reported the catalytic effect in the wet peroxide oxidation of formic acid and phenol [16]. The ceria catalyst has also been reported in the wet oxidation of pollutants [17,18]. However, only a few studies have been devoted to real industrial sludge. From the literature, we can conclude that Cu-loaded catalysts are good candidates for catalytic wet oxidation. Therefore, we want to perform a study on the performance of catalytic wet oxidation of pharmaceutical sludge over a Cu-based catalyst.

Volatile fatty acids (VFAs) could be produced considerably in the wet oxidation of sludge [19]. These acids include formic acid, acetic acid, propanoic acid, etc., which have potential for the utilization as organic carbon source in biological wastewater treatment plant. Gapes et al. reported that the yield of the acids could be produced by the wet oxidation of sludge [20]. Chung et al. [21] reported that propionic acid concentration increased up to 13.5 mg/L under the reaction temperature 240 °C in the wet oxidation process of sewage sludge. Wu et al. reported that propionic and butyric acid could be produced in the wet oxidation process of sludge [22]. As regards to VFAs production, the reaction temperature is the main factor that influences the degradation of sludge greatly. On the other hand, an additional amount of oxygen is a key parameter influencing acetic acid production [23]. Up to now, much available literature has reported studies on maximizing the production of acetic acid in the wet oxidation of sludge. Therefore, it can be concluded

that the production of acetic acid and VFAs in the wet oxidation of sludge is well worth studying.

In this study, the performance of catalytic wet oxidation of excess activated sludge derived from pharmaceutical wastewater treatment over a $Cu/\gamma\text{-}Al_2O_3$ catalyst was investigated. The effects of reaction conditions were examined, including additional catalyst amount, reaction temperature, time, and initial oxygen pressure. The production of acetic acids and VFAs was discussed. These results would provide important information for the wet oxidation treatment of sludge.

## 2. Materials and Methods

### 2.1. Materials

The present experiments were performed with pharmaceutical sludge, which was collected from a pharmaceutical factory (Zhejiang, China). The sludge was obtained from a thickening pond in the wastewater treatment plant and stored at 4 °C without further treatment. The characteristics of the sludge are as follows: TCOD 16,500~18,000 mg/L, TSS 16.3~17.8 g/L, VSS 13.2~14.1 g/L, pH 7.3~7.8. Because the sludge solution was not a homogeneous phase, these values have a certain range. Experimental reagents, such as $Cu(NO_3)_2$ and NaOH (AR), were purchased and used without any further treatment. These reagents were bought from Shanghai Sinopharm Chemical Reagent Co., Ltd., Shanghai, China.

### 2.2. Reaction System

Experiments were carried out in a SUS316 autoclave reactor [24]. The total inner volume of the reactor was 100 mL. For a typical experimental process, sludge solution was put into the reactor, with the volume 50 mL. Then the reactor was purged with 0.5 Mpa pure oxygen gas for three times. The oxygen gas was stored in the reactor by adjusting the pressure, as the oxidant. Then the reactor was heated up to desired reaction temperature from 180 °C to 260 °C. Once the reaction temperature was attained, a timer started counting from 20 to 60 min. The stirrer speed was set to 300 rpm/min. The pressure inside the reactor increased with the saturated vapor pressure and oxygen gas. When the reaction was finished, the reactor was removed immediately from the heating jacket, and cooled down to room temperature. The samples were collected and analyzed. The purity of the $O_2$ purchased from Huaxiong Gas Co., Ltd., Foshan, China, was 99.5%.

### 2.3. Analysis

The analysis indicators included TCOD, VSS, and pH. The analysis was carried out with standard methods [25]. VFAs were quantitatively analyzed using gas chromatography (GC, Persee G5, Shanghai, China) with bonded polyethylene glycol capillary columns (DB-FFAP, Agilent, Santa Clara, CA, USA) and a flame ionization detector (FID). Helium gas was used as the carrier gas, which were acquired from Huaxiong Gas Co., Ltd., Foshan, China. In the process of GC analysis, the temperature of injection, column oven, and detector were fixed at 220 °C, 160 °C, and 220 °C, respectively.

### 2.4. Preparation of Catalyst

The catalyst was prepared using a typical wet impregnation procedure. First, $Cu(NO_3)_2$ was dissolved in water with a concentration of 2.0 mol/L. The $\gamma\text{-}Al_2O_3$ carrier was immersed in the above solution, and its pH was adjusted to 10. The impregnation process continued for 24 h. After the impregnation process, the $\gamma\text{-}Al_2O_3$ solid was collected and heated for 4.0 h at 650 °C to obtain the $Cu/\gamma\text{-}Al_2O_3$ catalyst.

## 3. Results and Discussion

### 3.1. Characterization of the Catalyst

Figure 1 shows the SEM images of the prepared catalyst. As shown in Figure 1, it can be seen that many uniform particles landed on the surface of the $\gamma\text{-}Al_2O_3$. These particles

are closely arranged on the surface of the carrier of γ-Al$_2$O$_3$. Figure 2 shows the EDS spectra of the prepared catalyst. As shown in Figure 2, Cu was loaded successfully with a large amount. These analysis results ensure that the active component in the catalyst produces the catalytic effect.

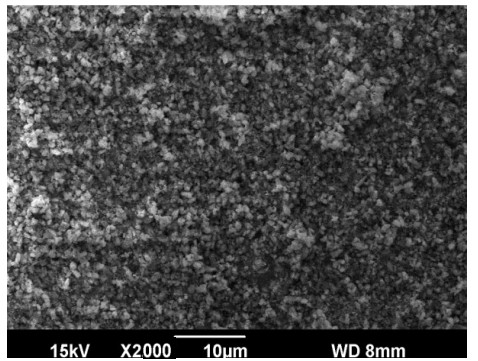 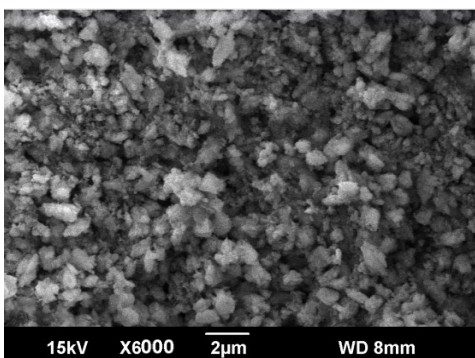

**Figure 1.** SEM images of the prepared catalyst.

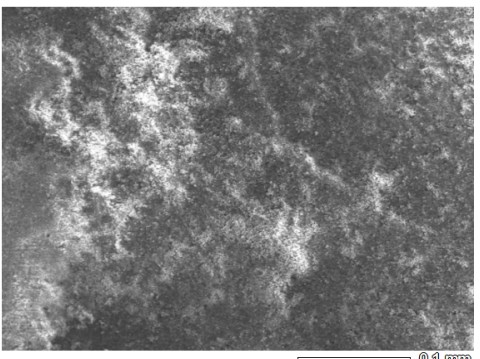 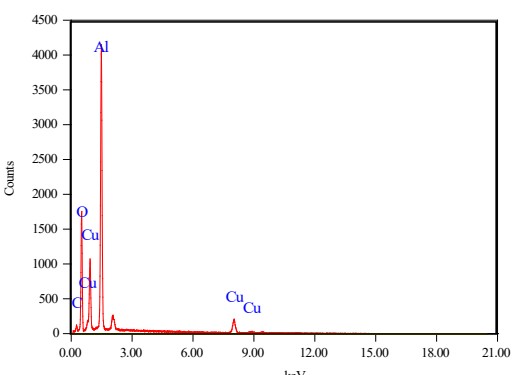

**Figure 2.** EDS spectra of the prepared catalyst.

### 3.2. Effect of Catalyst Amount

At first, experiments were carried out to assess the influence of the catalyst, with the catalyst addition amount from 0 to 7.0 g with the reaction temperature 260 °C, time 60 min, and initial oxygen pressure 1.0 MPa. TCOD and VSS removal rates were adopted to assess the catalytic efficiency of the catalyst. It was found that the removal rates of TCOD increased significantly with the increase of the amount of additional catalysts in Figure 3. The experimental results demonstrated that the fabricated catalyst had an excellent catalytic effect on the wet oxidation of organics in the liquid. However, the change in VSS removal is not big. The VSS removal was mainly from the dissolution and hydrolysis of sludge, which easily took place under high reaction temperatures. When the catalyst was added, the VSS content in the sludge only came into contact with the surface of the catalyst. Unlike COD removal, the catalyst enhanced the removal of pollutants due to its solubility. Therefore, the removal rate of VSS was up to 90% even though there was no additional catalyst. When the catalyst amount was added, the VSS removal rate only increased a little. However, to sum up, the additional catalyst can significantly enhance the catalytic wet oxidation of pharmaceutical sludge. With the additional amount of catalyst 5.0 g/L, the results were almost the best, compared with the results of 7.0 g/L. Therefore, we selected the additional amount of catalyst 5.0 g/L in the following experiments.

### 3.3. Effect of Reaction Temperature

A lot of literature has reported that the reaction temperature significantly influences the wet oxidation reaction. On one hand, the solubility of oxygen increases when the temperature is above 100 °C. On the other hand, according to Arrhenius's law, a higher

temperature would provide a higher reaction efficiency, which is favorable for the oxidizing reaction [26,27]. In this study, the influence of reaction temperature (in the range of 180–260 °C) was investigated.

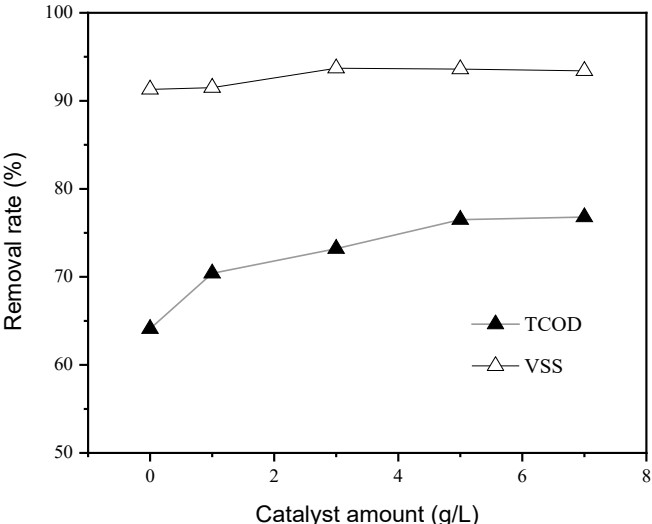

**Figure 3.** Effect of additional amount of catalyst on the TCOD and VSS removal (260 °C; 60 min; initial oxygen pressure: 1.0 MPa).

As shown in Figure 4, the increase of TCOD and VSS removal rates was very fast, according to Arrhenius's law in the kinetic regime. The TCOD removal rate was about 40% at 180 °C. Interestingly, it was about 75% at 260 °C. These results demonstrated that the reaction temperature significantly influences the wet oxidation process. Some carboxylic acids with small molecule weights were generated in the wet oxidation process. In particular, acetic acid was stable in the wet oxidation reaction. These results were similar to the observations of Strong [9], VSS removal rate 93% was acquired at 220 °C for 2 h under a pure oxygen pressure of 2.0 Mpa in the wet oxidation of sludge. Therefore, to obtain higher production of carboxylic acid, enough higher reaction temperature was necessary to increase the oxidation efficiency of the intermediates. On the other hand, surplus reaction temperature also increased the investment for the overall reaction equipment system. Therefore, it is necessary to optimize the reaction conditions considering the cost and production of useful chemicals.

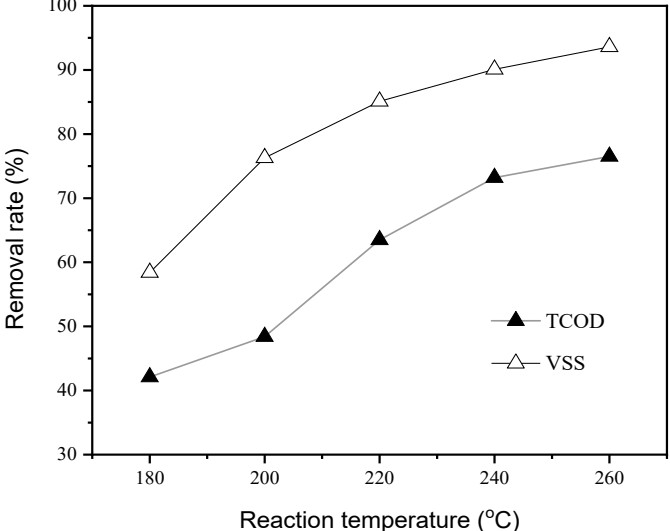

**Figure 4.** Effect of reaction temperature on the TCOD and VSS removal (60 min; initial oxygen pressure 1.0 MPa; catalyst 5.0 g/L).

### 3.4. Effect of Reaction Time

The influence of reaction time was discussed with the experimental results by varying the reaction time to 20~60 min. In Figure 5, the removal rates of TCOD and VSS increased together with the increase of reaction time. Interestingly, VSS removal took place with higher results even in a very short time, which means that hydrolysis very easily happened under higher reaction temperatures. A linear relationship between the increase in removal rates and reaction time existed. The possible reason may be that the cell wall was destroyed more completely under higher temperatures during the wet oxidation, which enhanced the solubilization, hydrolysis, and removal of TCOD and VSS. With the extension of time, the degradable organic materials accumulated in the liquid. Then, organic intermediates were oxidized further, which induced higher removal of TCOD. With respect to the balance between the efficiency and the cost for reaction time, the reaction time, 60 min, was chosen as the proposed reaction time.

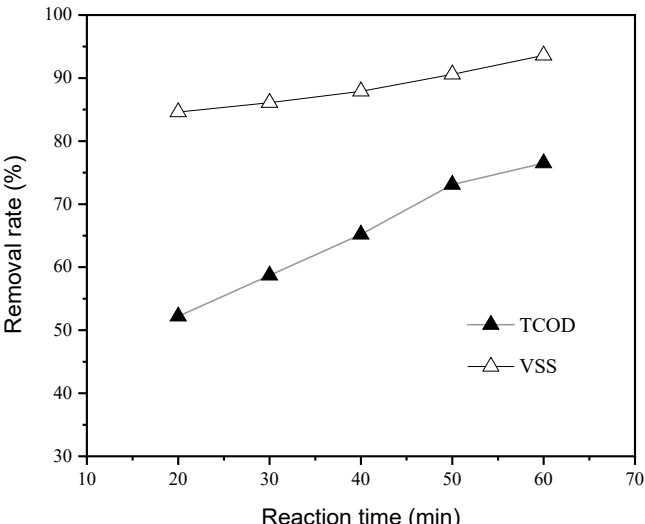

**Figure 5.** Effect of reaction time on the TCOD and VSS removal (260 °C; initial oxygen pressure 1.0 MPa; catalyst 5.0 g/L).

### 3.5. Effect of Initial Oxygen Pressure

Experiments were carried out with the initial oxygen pressures of 0.2~1.0 MPa to assess the influence of oxidant amount. As shown in Figure 6, an additional amount of oxygen plays an important role in the wet oxidation of pharmaceutical sludge. The TCOD removal rates increased significantly with the increase in the amount of oxygen. The reason may be that the increase of an additional amount of oxygen induced the increase of hydroxyl radicals, which possessed strong oxidation activity. On the other hand, the increase in oxygen pressure also enhanced the dissolution of oxygen in the liquid, which increased the reaction efficiency. By comparison, it can be seen that the increase in oxygen amount mainly increased the TCOD removal, and the VSS removal increased a little. When the additional amount of oxygen was 1.0 MPa, the highest TCOD removal rate was obtained. The possible reason may be that some non-oxidizable intermediates were generated during the wet oxidation process. For example, acetic acid is more stable under hydrothermal conditions [28]. These experimental results showed that adding oxygen is enough for the wet oxidation of pharmaceutical sludge.

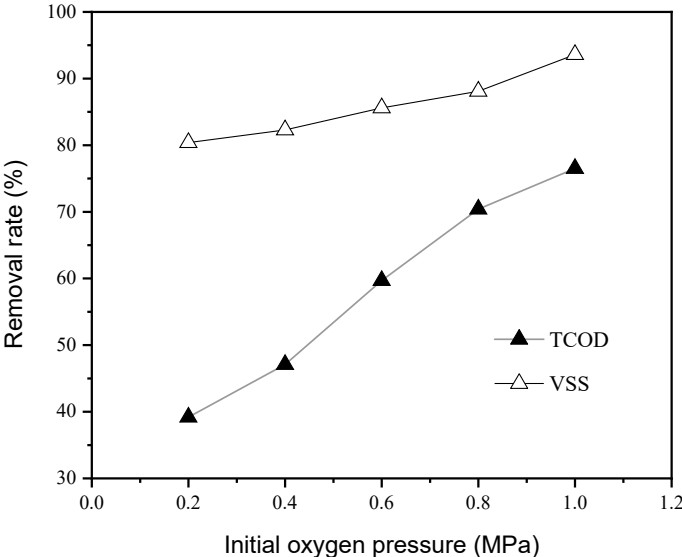

**Figure 6.** Effect of initial oxygen pressure on the TCOD and VSS removal (260 °C; 60 min; catalyst 5.0 g/L).

### 3.6. Production of VFAs

During the wet oxidation reaction, organic pollutants could be oxidized to small molecules products, such as organic acid, even carbon dioxide and $H_2O$. VFAs are the dominant components, which were always considered as value-added products. Figure 7 shows the effect of reaction temperature on the concentration of acetic acid and VFAs concentration at 260 °C for 60 min and initial oxygen pressure of 1.0 MPa, catalyst amount 5.0 g. In Figure 7, the concentration of VFAs was calculated as the sum of the concentrations of acetic, propanoic, isobutyric, and isovaleric acid. It was found that acetic acid was the main carboxylic acid of VFAs. VFAs concentration increased dramatically while the reaction temperature rose. The reaction temperature of 180 °C is too low for the production of VFAs. However, there is little change when the reaction temperature increases from 240 to 260 °C. Chung et al. obtained similar data with the reaction temperature from 180 °C to 240°C [21]. Shanableh et al. found that there was a limit to the production of VFAs from the wet oxidation process since excess oxidants can stimulate oxidation [29]. Because acetic acid is a stable intermediate and was relatively hard to be oxidized under hydrothermal conditions, acetic acid dominated the intermediates after the wet oxidation. The $BOD_5$/COD value of the liquid after the wet oxidation was above 0.70, which means that the carboxylic acids in the liquid were easy to adopt for the survival of microorganisms. In the industrial wastewater treatment process, sodium acetate was always bought as an organic carbon source. Therefore, the liquid of wet oxidation could be used as an organic carbon source, saving the acetate sodium cost. Then, once the VFAs are used as an organic carbon source for the biological treatment of wastewater, the cost of the wet oxidation technology would be decreased. Thus, the economic cost of wet oxidation would be more easily accepted. In fact, the VFAs produced by the wet oxidation of sludge include more than ten carboxylic acids. Future work should be performed to investigate the composition in detail.

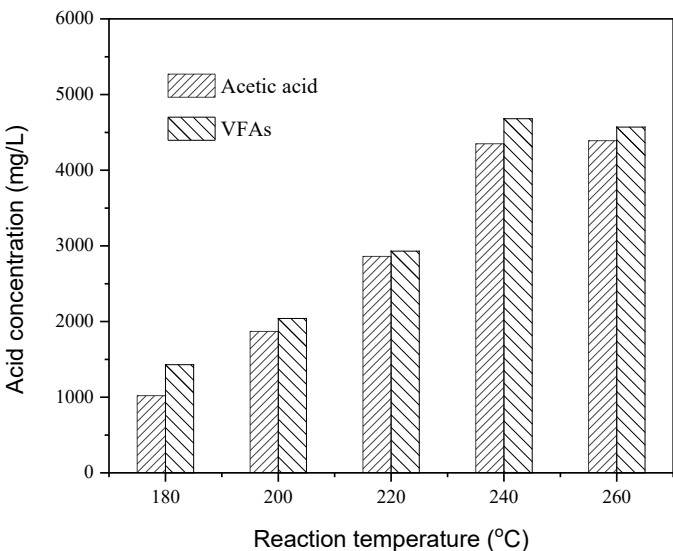

**Figure 7.** Effect of reaction temperature on the production of acetic acid and VFAs (260 °C; 60 min; catalyst 5.0 g/L; initial oxygen pressure 1.0 MPa).

*3.7. Discussion*

At present, the detailed mechanism of sludge wet oxidation is still unclear because sludge composition is very complicated, and many different materials were generated in the degradation of sludge and its hydrolysis products. As Urrea reported, wet oxidation is considered a process because of the hydroxyl radicals [30,31]. In the hydrolysis process, the efficient deconstruction of sludge happened, which induced the dissolution of proteins and lipids into the liquid. Thus, a high VSS removal rate was obtained. It should be noted that the hydrolysis process performed very quickly under high reaction temperature. After that, the intermediates were oxidized by the strong oxidant free radicals, producing a large amount of VFAs, such as acetic acid, isobutyric acid, propanoic acid, etc. The $BOD_5/COD$ ratio of the liquid was very high, even above 0.70, which demonstrated that the liquid generated after the wet oxidation could be used as an organic carbon source for the biological treatment process of wastewater because the microorganism easily adopted it. In our previous study, we performed a related study on the prepared $CuO-CeO_2/\gamma-Al_2O_3$ catalyst [32]. In the present study, the COD removal rate was enhanced with the addition of a prepared catalyst, which means that the pollutants in the solution were oxidized further. Based on the discussion on the reaction parameters, the production of acetic acid and VFAs was analyzed, verifying the prepared catalyst's effect. The Cu of the catalyst was oxidized form, which was produced from $Cu(OH)_2$ in the calcination process. Based on the catalytic activity of Cu, the wet oxidation of organic pollutants in the solution was enhanced, and the $\gamma-Al_2O_3$ was selected as a carrier. The reason for selecting an immobilized catalyst is that it avoids the loss of a reactive catalyst in the wet oxidation process. In some industries, acetate sodium was always bought to increase the biological treatment efficiency. Therefore, the liquid could be used for the biological treatment of wastewater. On the other hand, the wet oxidation belongs to exothermic reaction, which could be used to amend the energy consumption for the maintainence of the reaction. Furthermore, the inorganic particles of sludge became smaller, which was favorable to release intercellular water and increased the dehydration performance. The inorganic solid could be used for the production of permeable brick. Thus, wet oxidation technology could be selected as a suitable technology for the treatment of sludge.

It has been reported that the residual toxicity might be removed during treatment at the wastewater treatment plant. Therefore, the toxicity of pharmaceutical sludge decreased significantly after the wet oxidation. Wet oxidation could be used as a primary treatment method for hazardous pharmaceutical sludge. Future research should focus on identifying and quantifying intermediates, and a suitable catalyst for producing value-added chemicals



should be developed further. In a future study, an excellent catalyst with fast reaction efficiency and lower reaction temperature should be developed preferentially. Simultaneously, the detailed reaction mechanism is expected to be investigated further. Furthermore, it should be noted that the reaction devices were very important for industrial utilization.

## 4. Conclusions

The present study investigated the performance of catalytic wet oxidation of excess activated pharmaceutical sludge over a $Cu/\gamma\text{-}Al_2O_3$ catalyst. The experiments were performed with a stainless steel autoclave reactor by using the prepared $Cu/\gamma\text{-}Al_2O_3$ catalyst. The effects of reaction conditions were examined, including additional catalyst amount, reaction temperature, time, and initial oxygen pressure. Results demonstrated that the catalyst, fabricated via a co-precipitating method, has excellent catalytic performance. Through the study on condition optimization, the highest removal rates of VSS (93.6%) and TCOD (76.5%) were acquired with the reaction temperature 260 °C, time 60 min, initial oxygen pressure 1.0 MPa, and the prepared catalyst 5.0 g/L. The volatile fatty acids (VFAs) produced from the wet oxidation included acetic (mainly), propanoic, isobutyric, and isovaleric acids, which have great potential for the utilization as organic carbon sources in biological wastewater treatment plant. These results demonstrated that the proposed method, catalytic wet oxidation over a $Cu/\gamma\text{-}Al_2O_3$ catalyst, is effective for treating excess activated sludge and resource utilization of organic carbon in the sludge.

**Author Contributions:** X.Z. and H.L. contributed to the conception. Material preparation, experiments conduction, data collection, and analysis were performed by S.C. and Y.Z. The first draft of the manuscript was written by S.C. The manuscript was edited and revised by X.Z. and H.L. All authors have read and agreed to the published version of the manuscript.

**Funding:** The authors gratefully acknowledge the support from the National Natural Science Foundation of China (51978499).

**Data Availability Statement:** Data will be made available upon request.

**Conflicts of Interest:** The authors declare no conflict of interest.

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
