# Peer review of "High-Performance Catalytic Wet Oxidation of Excess Activated Sludge Derived from Pharmaceutical Wastewater Treatment Process over a Cu/γ-Al2O3 Catalyst"

_water, doi:10.3390/w15193494_

Round 1
Reviewer 1 Report
The manuscript entitled ‘High-performance Catalytic Wet Oxidation of Excess Activated Sludge Derived from Pharmaceutical Wastewater Treatment over Cu/γ-Al2O3 Catalyst’ is submitted in WATER by Shangye Chu et al is fascinating work. Here authors explain and demonstrate the sort of method for pharmaceutical sludge issues to convert it into useful chemicals by the Wet Oxidation process. I have the following suggestions and queries for the work.
Line 91-92: Chung et al. [19] reported that propionic acid concentration increased from 0 mg/L at 180 °C to 13.5 mg/L at 240 °C in the wet oxidation of sewage sludge. Here, the authors understand 0 mg/L at 180 °C ? Please cross-check.
Line 170-172: Therefore, the removal rate of VSS was up to 90% even though there was no additional catalyst. When the catalyst amount was added, the increase of VSS removal rate only increased a little. The authors should explain the reason, why after increasing the amount of catalyst VSS % increases little as shown In Fig-3.
Figure 4: Authors choose a high temperature of 260 0C as the maximum. I believe, that if they touch 300 oC TCOD may enhance or decrease. So here, temperature is not optimized for the effect. In Figure 5 time was optimized 60 minutes it should also check at a higher time zone for VSS and TCOD.
The authors should reach a plausible mechanism for the Cu/γ-Al2O3 catalyst as per the wet-oxidation process.
The authors explain in detail that Cu-based systems are well-performing systems for the wet-chemical process. What is the role of γ-Al2O3? Is Cu (reduced form) attached to γ-Al2O3 or it contain the Cu+ ions?
Overall, the manuscript is well written and I shall recommend minor revision after responding to the above-mentioned queries.
Reviewer 2 Report
The authors investigated an important problem of pharmaceutical wastewater utilization by wet oxidation using Cu/gamma-Al2O3 catalyst. The proposed method came out to be effective for the treatment of excess activated sludge and the organic compounds in the sludge could play a role of organic carbon resource. However, English of the all manuscript should be corrected extensively. It is hardly possible to read. The overall value of the manuscript is reduced due to inapropriate language.
Some mistakes to be corrected and remarks to be answered are listed below:
p.2, line 83 It is: pollants; should be: pollutants;
p.3, line 113 All the shortcuts should be explained.
p. 3, line 114 It is: were purchased without any farther treatment; should be were purchased and used without any farther treatment;
p.3, line 122 ...and the pressure was increased to na desired value.
p.3, line 124 Rewrite the sentence...The reaction started and was continued for 60 min.
p.3, line 127 The pressure inside the reactor increased due to the rose of the saturated vapor pressure during the reaction.
p.3, line 130 ...was then be collected
p.3, line 131 The purity of the O2 purchased from..., was 99,5%.
p.3, line 134 Rewrite the sentence. It is grammatically incorrect.
p.3, line 136 Instead of ‘quantitative’ put ‘quantitatively’.
p.3, line 140 The temperature of injection
p.3, line 141 Remove space between ‘220’ and ‘°C’.
p.3, line 142 The gamma-Al2O3 carrier was immersed in the above solution and its pH was adjusted to 10. The impregnation process was continued for 24h.
p.4, line 146 Instead of ‘baked’ should be ‘heated’.
p.4, line 173 Usually, the authors refer in the ‘Results and Discussion’ part to a particular result, presented in a Figure or Table and draw conclusion. Which experimental result do confirm excellent catalytic effect?
Correct all this part (Res. and Disc.) taking this suggestion into account.
p.6, line 196 AA was more reactive than what?
p.6, lines 207-208 The sentence was not finished.
p.7, line 259 What does the B/C value of the liquid mean?
p.7, line 261 It is: acetate sodium; should be: sodium acetate;
p.8, line 266 remove ‘to be’
p.8, line 267 remove ‘types of’
p.8, line 277 Instead of ‘disolve’ write ‘disolution’
p.8, line 283 to the end of the ‘Results and Discussion’ part
This part contains too many general statements. Focus on what exactly this work improved/changed.
p.8, line 290-292 Rewrite this sentence. Which pores do increase? Have you any prove for that?
p.9, line 303 What does the expression: ‘lower reaction conditions’ mean?
p.9, line 310 Remove ‘prepare’.

English of the all manuscript should be corrected extensively.
Reviewer 3 Report
In this work, the authors prepared and characterized Cu/γ-Al2O3 catalyst by wet impregnation procedure, evaluating the effect of the catalyst amount, reaction temperature, time and initial oxygen pressure for the treatment of excess activated sludge derived from pharmaceutical wastewater treatment. Thus, it can be recommended for publication in the journal of “Water” after the following addressed:
1. Abstract part: To show details about the characterization of the Cu/γ-Al2O3 catalyst and quantitative results obtained in this manuscript (such as characterization and photodegradation kinetics).
2. Introduction part:
(a) Comment about the heterogeneous photocatalysis such as advanced oxidation process (AOPs) for the photodegradation. For example:
- Journal of Molecular Liquids (https://doi.org/10.1016/j.molliq.2022.121090)
- Separation and Purification Technology (https://doi.org/10.1016/j.seppur.2019.01.066)
3. Materials and Methods / Results and Discussion
(a) Present an illustrative figure about the synthesis of Cu/γ-Al2O3 catalyst;
(b) All results need in-depth discussion and correlation with the literature;
(c) Show better statistical detail about the ideal condition of the photocatalysis process, such as compound central rotational design (CCRD), as well as correlations between photocatalytic activity and characterization results, such as SPSS Statistics;
(d) Evaluate the effect of the presence of radical scavengers and reuse;
(e) Carry out photocatalytic tests under visible radiation and purchase with commercial catalysts;
(f) Present an item on statistical analysis and methodological details on the characterization. Also, present results from more characterization techniques, such as XRD, DRS, zeta potential, pHZCP and FTIR.
Round 2
Reviewer 2 Report
I recommended extensive edition of English. The authors only corrected most of errors indicated and didn't answer some question. I recommend rejection now.
There are still a lot of mistakes, for example:
line 294 (2nd version) The existence of Cu in the catalyst was the oxidized form;
line 303 (2nd version) ...which is conductive...
line 312 (2nd version) Two first sentences are not grammatically correct;
and many others.
I recommended extensive edition of English. The authors only corrected most of errors indicated and didn't answer some question. I recommend rejection now.
There are still a lot of mistakes, for example:
line 294 (2nd version) The existence of Cu in the catalyst was the oxidized form;
line 303 (2nd version) ...which is conductive...
line 312 (2nd version) Two first sentences are not grammatically correct;
and many others.
Author Response
Thank you very much for your kind comments. In accordance with your comments, we revised the manuscript carefully. Hopefully, the manuscript is much better now.
Thanks again!
Reviewer 3 Report
All the comments were addressed and the manuscript can be accepted in its current form
Author Response
Thank you very much for your comments.
Round 3
Reviewer 2 Report
-